# Effect of Number of Household Members on Falls among Disabled Older People

**DOI:** 10.3390/ijerph19105888

**Published:** 2022-05-12

**Authors:** Juyeong Kim, Ye Seol Lee, Tae Hyun Kim

**Affiliations:** 1Department of Public Health, Sahmyook University, Seoul 01795, Korea; jennajyk394@gmail.com; 2Department of Rehabilitation Medicine, Seoul National University Hospital, Seoul 03080, Korea; yeseol.lee127@gmail.com; 3National Traffic Injury Rehabilitation Research Institute, National Traffic Injury Rehabilitation Hospital, Yangpyeong 12564, Korea; 4Department of Hospital Administration, Graduate School of Public Health, Yonsei University, Seoul 03722, Korea

**Keywords:** living arrangements, falls, hip fracture, disability, cognitive decline

## Abstract

Objectives: To investigate the effect of the number of cohabitating household members on falls among an disabled aging Korean population. Methods: We used data from the first to the fourth waves of the Korea Longitudinal Study of Aging. Using the first wave at baseline, data included 1414 individuals aged 45 years and older who needed assistance for performance of activities of daily living (ADL) or instrumental activities of daily living (IADL). We classified falls as overall falls, falls requiring medical treatment, and hip fractures caused by falls. The number of cohabitating family members was classified as none (living alone), one, two, or more. A generalized estimating equation with logit link was used to examine the association between the number of cohabitating household members with overall falls and injuries caused by falls. Results: Compared to living with two or more household members, living alone was associated with higher odds of overall falls, falls needing medical treatment, and hip fractures caused by falls (odds ratio (OR) 2.13, 95% confidence interval [CI] 1.36–3.34; OR 2.13, 95% CI 1.28–3.53; OR 1.93, 95% CI 1.01–3.69, respectively). These associations were particularly strong in individuals with cognitive decline. Conclusions Living alone is associated with higher odds of overall falls, falls needing medical treatment, and hip fractures caused by falls, particularly for those with cognitive decline. Conclusions: Intervention programs to prevent falls in disabled, aging adults, especially those living alone and those with declined cognitive function, need to provide home care services and promote the use of safety equipment.

## 1. Introduction

Korea is one of the most developed countries in the world, with a rapidly aging population. The percentage of older adults in South Korea is currently 9%, and is projected to reach 38% by 2050, ranking it as one of the countries with the highest percentage of elderly population [1]. Falls are a major cause of disability, healthcare utilization, and mortality in older populations [2]. Fall-related deaths in older persons have been on the rise in the past 15 years, and the incidence of injuries due to falls among older adults, aged 65 years or older, is estimated to be 87%. Risk factors for falls include balance deficits, living arrangements, history of previous falls, difficulties with ambulation, visual impairments, age, gender, multiple risk factors, and certain chronic diseases [3]. Falls can be caused by interactions between multiple risk factors [4]. Given the negative effects of falls on the physical and mental health of older adults, it is imperative for healthcare practitioners to identify factors leading to falls and develop ways to prevent community-dwelling older adults from falling [4].

More than half of older adults experience falls at home where they tend to spend the majority of their time at [5]. The aging population living at home with physically disabling conditions is found to be more likely at a high risk for both, falls and hip fractures, which is one of the most devastating and expensive problems faced by older adults; it commonly results in a permanent disability, nursing home placement, or death [6,7]. A previous study reported that falls occurred while engaging in activities such as slipping, walking, transferring, and going up or down the stairs [8], and during activities such as eating, cooking, and resting alone [9]. However, it is difficult for older persons with disabilities to perform these activities alone and they need assistance from family members. Therefore, focusing on characteristics related to their living arrangements, such as the number or composition of household, is important to understand and plan for the proper care of these individuals [10].

We hypothesize that the number of household members residing with older adults might be associated with the incidence of falls. A larger number of family members, living with older adults with functional disabilities, is found to be associated with better instrumental and social support [11]; in addition, the amount of care, both formal and informal, received from cohabitating household members, is minimal or unavailable to older adults living alone [12,13]. In addition, the number of household members is known to reduce the survival and functional decline of older populations [10]. Therefore, the association between the number of household members and falls among older adults with disabilities, living at home, should be investigated. Several studies have examined the association between the number of household members and falls among older people. A previous study conducted in Australia showed an increased risk of fall among those living alone compared to those living with someone, although this was not discussed [14]. Older adults living alone were found to have more falls than those living alone in China [15]. Another study found that people living alone were more likely to be associated with falls than those living with their family members or with people outside their family, in community-dwellings middle-aged and older adults in rural areas of the United States [16]. However, previous studies focused mostly on the status of cohabitation; studies on the association between the incidence of falls and the number of cohabitating family members have not been reported. In addition, the presence of a household member can be important for older persons with a disability to reduce falls, and injuries caused due to falls. Studies examining the association between number of household members and falls, and injuries caused due to falls are insufficient, especially those on the disabled older population. Finally, despite the relevance of this issue, studies on this topic have not been conducted in Korea, which is a developed country with a rapidly aging population. It is necessary to study the relationship between the number of household members and the incidence of falls among senior citizens with disabilities. This will enable identification of risk factors for fall management of elderly adults with disabilities who have difficulties in performing daily life activities; however, no such study has been reported in literature, yet. Therefore, the identification and removal of risk factors associated with falls are of high clinical priority.

Thus, the first aim of this study was to investigate the effects of the number of cohabitating household members on overall falls and injuries related to falls, including hip fractures and others needing medical treatment, in the disabled aging Korean population. Based on a previous study, the present study hypothesized that a smaller number of family members who cohabitate in a household with disabled older persons is associated with higher odds of the older person experiencing overall falls, falls needing medical treatment, and hip fractures caused by falls. In addition, the incidence of falls is high among elderly people with physical and psychological disabilities. According to a literature review, the inability to perform basic and instrumental activities of daily living is associated with a faster rate of decline in the cognitive functioning of older adults [17], which in turn was identified as a risk factor for falls among older adults [18]. Considering that the presence of cognitive dysfunction would affect fall risk, research on the association between the number of household members and falls according to cognitive status needs to be discussed. Therefore, we additionally investigated the effects of the number of household members on the incidence of overall falls, falls needing medical treatment, and hip fractures caused by falls according to the cognitive status of the elderly participants.

## 2. Methods

### 2.1. Data

Data were obtained from the 2008, 2010, and 2012 waves of the Korea Longitudinal Study of Aging (KLoSA), which is a nationally representative study of non-institutionalized South Korean adults across 15 large administrative areas; all data are available via a national public database (https://survey.keis.or.kr/klosa/klosadownload/List.jsp) [19]. 

Assuming a target sample size of 10,000 persons and an average household size of 1.67 members aged ≥45 years, a total of 1000 enumeration districts were selected and stratified by area (urban/rural) and housing (apartment/non-apartment). The first wave of the survey, conducted between July and December 2006, involved 10,254 older adults (aged ≥45 years) in 6171 households. All participants were interviewed face-to-face, using a computer-assisted personal interviewing method. In the second wave of the survey, conducted between July and November 2008, re-interviews were conducted with 8688 (84.7%) respondents from the first wave. The third wave of the survey, conducted in 2010, included 7920 (80.3%) respondents from the first wave. Finally, 7486 (76.2 %) respondents from the original panel were interviewed in the fourth wave of the survey.

### 2.2. Study Population

This study targeted individuals who needed assistance for performance of activities of daily living (ADL) or instrumental activities of daily living (IADL). Therefore, 1516 of the 10,254 participants included in the 2006 survey (baseline year) and those who did not need any help for either ADL or IADL (n = 8738) were excluded from the study population. In addition, we only included participants who had all the information required for this study. Therefore, we excluded participants with missing body mass index (BMI) (n = 98) and care hours (n = 4). Thus, the final sample for 2006 comprised 1414 observations. The number of observations in the follow-up years was as follows: 1063 in the 2008 survey, 995 in the 2010 survey, and 845 in the 2012 survey (Figure 1).

### 2.3. Study Variables

#### Overall Falls, Falls Needing Medical Treatment, and Hip Fractures Caused by Falls

For outcome variables, overall falls, falls needing medical treatment, and hip fractures caused by falls were used. Each outcome was assessed using self-reported binary questionnaires with ‘yes’ or ‘no’ responses. The incidence of falls was assessed via a question that asked whether the participants had experienced falling in the past two years or not. Falls requiring treatment, were assessed via a question that asked whether the participants had recently experienced severe falls requiring medical treatment. Hip fracture caused by falls was assessed via a question that asked whether the participants had a hip fracture due to falling.

### 2.4. Number of Household Members

The number of cohabitating family members in the household was assessed using self-reported questionnaires. It was classified as none (living alone), one, two, or more.

### 2.5. Control Variables

To assess the independent influence of the dependent variables, we considered possible covariates among the various sociodemographic and health-related variables. Sociodemographic variables included age, sex, educational level, and equivalent household income. Age was divided into four categories: 45–64 years, 65–75 years, 76–85 years, and >85 years. The patients were divided according to sex. Educational level was divided into four categories: lower than elementary school, middle school, high school, and college graduate or higher. Equivalent household income was calculated as the total household income divided by the square root of the number of household members. These scores were then divided into quartiles: ≤25th percentile, 25th–50th percentile, 50th–75th percentile, and 75th–100th percentile. Care hours was defined as the number of hours during which care was provided to the participants by their primary caregiver and was used as a continuous variable. Health-related variables included IADL, BMI, Mini-Mental State Examination (MMSE) scores, number of chronic diseases (0, 1, and ≥2), and regular exercise (yes/no). Functional status was measured using the 10-item Korean Instrumental Activities of Daily Living Scale, which includes items on personal grooming, excursions for short distances, transportation use, making and receiving phone calls, managing money, performing household chores, preparing meals, shopping, taking medications, and doing laundry. During the health examination survey, height and weight were directly measured when the participants wore light clothing and no shoes. Height was measured to an accuracy of 1 cm, and weight to 0.1 kg. For each participant, BMI was calculated as the weight in kilograms divided by the square of the height in meters. The MMSE was used to measure cognitive function using the Korean MMSE, with a total score ranging from 0 to 30. The validity of the test was confirmed in a previous study [20].

### 2.6. Statistical Analysis

We used the chi-square test for binary variables and the Wilcoxon Mann–Whitney U test for ordinal variables to calculate the frequency and proportion of each variable. We ran a generalized estimating equation (GEE) owing to its flexible approach for analyzing correlated data from the same individuals over time [21]. The outcomes in our study were falls, falls needing medical treatment, and hip fractures caused by falls, all of which were binary. Therefore, three GEEs with a logit link were conducted separately to test the association between the number of cohabitating family members in the household and each outcome. To determine whether the probability of these three outcomes changed over time, we included time (years) as a categorical covariate in the model. Covariates of interest from all participants were added to the model to determine their effects on the probability of reporting any changes in the incidence of falls, falls needing medical treatment, and falls with hip fracture and to determine the probability of changes in the dependent and independent variables annually (Figure 2). For all analyses, a *p* value of 0.05 or lower was considered statistically significant. All analyses were conducted using the Statistical Analysis Software, SAS version 9.4 (SAS Institute Inc., Cary, NC, USA).

## 3. Results

Table 1 presents the general characteristics of the covariates included in this study at baseline. Data from 1414 respondents were included in the baseline year 2006. When broken down according to the number of family members in the household (n = 1414), 127 participants lived alone, 540 cohabitated with one member, and 747 cohabitated with two or more members.

Table 2 shows the adjusted effect of the number of members in the household on overall falls, falls needing medical treatment, and hip fractures caused by falls. Older adults living alone and those living with one person had higher odds of experiencing overall falls (odds ratio (OR) 2.13, 95% confidence interval [CI] 1.36–3.34; OR 1.54, 95% CI 1.08–2.19, respectively) than those living with two or more family members. In addition, those living alone had higher odds of experiencing both falls needing medical treatment (OR 2.13, 95% CI 1.28–3.53) and hip fractures caused by falls (OR 1.93, 95% CI 1.01–3.69) than those living with two or more family members.

Table 3 outlines the adjusted stratified analysis of the association between the number of cohabitating family members in the household and overall falls, falls needing treatment, and hip fracture caused by falls according to the cognitive function of the participants. Compared to participants with normal cognitive function, those with cognitive decline (MMSE score ≥ 18) exhibited higher odds of experiencing overall falls (OR 2.08; 95% CI 1.27–3.38), falls needing medical treatment (OR 2.15; 95% CI 1.24–3.72), and hip fracture caused by falls (OR 2.15; 95% CI 1.05–4.43), when living alone.

## 4. Discussion

This study aimed to identify the association between the number of cohabitating family members in a household and the incidence of overall falls, falls needing medical treatment, and hip fractures caused by falls among the disabled elderly Korean population. Our results indicate that living alone is associated with higher odds of falls, falls needing medical treatment, and hip fractures caused by falls, particularly for participants with cognitive decline.

These findings are consistent with those of a US study based on a community-dwelling population of middle-aged and older adults in a rural area [16], and with studies conducted in the UK [13] and Australia [14]; these studies reported that older persons living alone exhibited an increased risk of falls than those living with people. On the other hand, a study based on Chinese community-dwelling older adults in Hong Kong suggested that living with others may not be as safe as we assume [22]. The present study targeted people with disabilities as they are more vulnerable to falls and injuries [18]. Furthermore, the present study examined the incidence of overall falls, injuries caused by falls, including falls needing medical treatment and hip fractures caused by falls.

The association between the number of household members and overall falls, falls requiring medical treatment, and hip fractures occurring due to falls can be explained by several possible mechanisms. Our study revealed that, compared to living with two or more family members, living alone was associated with higher odds of overall falls, falls needing medical treatment, and hip fractures caused by falls. Participants living alone lacked social support in terms of informal care and living arrangements, which have been reported to be associated with falls [23]. Studies have also reported that elderly people living alone were more likely to live with lower social support and higher use of community health services (i.e., home health providers, social workers, case managers, aides) [24,25]. Considering that disabled elderly people need more social support even for basic activities of daily life, those living alone would be exposed to falls and injuries due to falls more often than those living with two or more household members. Additionally, a relatively high poverty rate while living alone could affect access to safe equipment to prevent falls. Older adults who live alone are more likely to live in poverty (18.6%) later in life than those living with family members (5.8%) [24]. Furthermore, home modifications can reduce the incidence of injuries caused by falls, even at low cost [26]. However, for those who are poor and live alone, access to safety equipment for making home modifications and safer outdoor activities may be low due to the relatively low affordability of such equipment [27].

In addition, the effect of living alone on the experience of overall falls, falls needing medical treatment, and hip fractures caused by falls was only significant and slightly larger among those with cognitive decline. According to a previous study, a decline in cognitive function increases the risk of falls [28]. It has been reported that older people with cognitive impairment have slower walking speed, poorer obstacle clearance, impaired coordination and balance, reduced balance control when performing a secondary task, and a greater likelihood of developing mobility problems [29,30,31]. The neurodegenerative process associated with dementia may also increase the risk of falls by increasing the likelihood of autonomic dysfunction, including symptomatic orthostatic hypotension [32]. Finally, fall risk may be increased by the use of psychotropic medications, particularly sedative hypnotics, antipsychotics, and antidepressants [33], which are more commonly prescribed to people with cognitive disorders. Considering that disabled elderly people are already vulnerable to falls, the subgroup of participants with cognitive decline who live alone have higher odds of experiencing overall falls, falls needing medical treatment, and hip fracture caused by falls.

A major strength of our study is its novel approach for investigating the impact of different numbers of household members on falls and injuries caused by falls in the disabled elderly Korean population. Our findings emphasize the need to focus on providing support for the prevention and treatment of falls among disabled elderly people living alone, especially for those with cognitive decline. Additionally, this study was performed based on a nationally representative aging Korean population

However, this study has some limitations. First, the longitudinal design used in this study diminishes our ability to determine the causal relationship between the number of household members and falls and injuries caused by falls. In other words, our results could reflect a causal relationship contrary to that ascribed to the relationship between the number of household members and falls and injuries caused by falls (e.g., the experience of falls or injuries due to falls leads to living alone). Future studies should investigate the effect of changes in the number of household members on health outcomes or the effect of the same on later health outcomes. Second, the respondents’ reports were subjective and imperfect measures, potentially affected by perception bias and adaptation of resources. In addition, the association between the number of household members and falls can differ according to older adults’ residential places, such as homes and institutions. However, this study only focused on those people living at home because the KLoSA survey targeted only those individuals living at home. Further studies on this topic must compare these findings to the responses of participants living in an institution. Finally, due to the observational nature of this study, it cannot completely eliminate the possibility of residual confounding, such as receiving geriatric care and other potential sources of bias.

Our findings have several implications for policy. Considering that participants living alone were identified as a vulnerable group for falls, more effective and attentive fall prevention interventions should be designed and implemented to lessen the economic and human burden, especially for the disabled elderly population. Elderly people living alone should be given more attention as they are a population in need of care. Often, those living alone receive less supportive care and are more likely to use one or more community health services (i.e., home health providers, social workers, case managers, and aides) [25]. Therefore, health care services should be adjusted to provide a form of home visiting health care services for the disabled elderly living alone to lessen their risk of falls that may occur while travelling from their home to the clinic and daily living. At the same time, it will also be necessary to encourage those living alone who do not use the available services to use care. Additionally, educational programs are needed for disabled adults to teach them how to maintain balance or fulfill tasks (i.e., putting on their pants while changing clothes) through balance training, instruction on safety and adaptive equipment, compensatory techniques to help improve safety and reduce the risk of falls; schemes to provide financial support to acquire safety equipment may also be considered [34].

## 5. Conclusions

Our results indicate that living alone is associated with higher odds of overall falls, falls needing medical treatment, and hip fractures caused by falls, particularly in participants with cognitive decline. Considering that participants living alone were identified as a vulnerable group for falls, a more effective fall prevention intervention should be designed. In particular, government support for prevention and management of falls in disabled older adults living alone, in the form of home visiting care services and promotion of use of safety equipment, is required.

## Figures and Tables

**Figure 1 ijerph-19-05888-f001:**
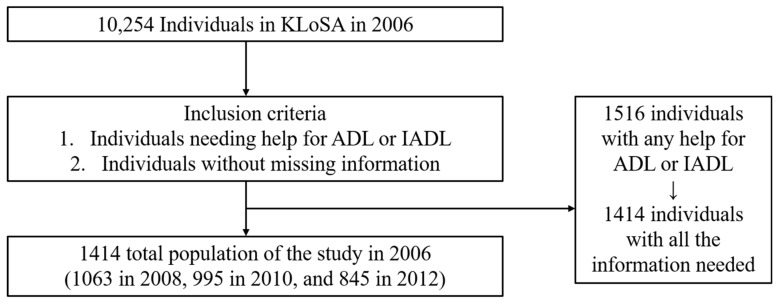
Flow chart of selection of the study participants.

**Figure 2 ijerph-19-05888-f002:**
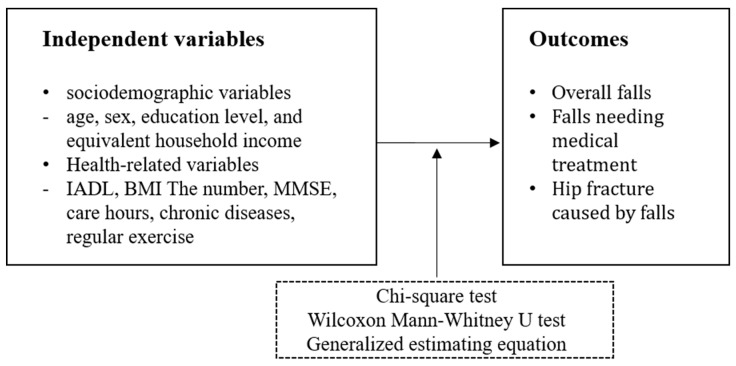
Variables and Statistics.

**Table 1 ijerph-19-05888-t001:** General characteristics by incidence of overall falls, falls needing medical treatment, and hip fractures caused by falls.

Variable	N	Overall Falls (n = 1414)					Falls Needing Medical Treatment (n = 1414)					Hip fractures caused by Falls (n = 1414)				
		Yes		No			Yes		No			Yes		No		
		N	%	N	%	*p*-Value	N	%	N	%	*p*-Value	N	%	N	%	*p*-Value
**Total**	1414	96	6.8	1318	93.2		67	4.7	1347	95.3		28	2.0	1377	97.4	
**Number of household members ^a^**																
0 (single household)	127	16	12.6	111	87.4	0.021	14	11.0	113	89.0	0.002	8	6.3	119	93.7	0.022
1	540	36	6.7	504	93.3		23	4.3	517	95.7		11	2.0	529	98.0	
2≤	747	44	5.9	703	94.1		30	4.0	717	96.0		18	2.4	729	97.6	
**Age**																
45–64	487	18	3.7	469	96.3	0.007	15	3.1	472	96.9	0.174	9	1.9	478	98.2	0.620
≤75	426	37	8.7	389	91.3		26	6.1	400	93.9		13	3.1	413	97.0	
≤85	379	33	8.7	346	91.3		20	5.3	359	94.7		11	2.9	368	97.1	
85<	122	8	6.6	114	93.4		6	4.9	116	95.1		4	3.3	118	96.7	
Sex																
Male	718	26	3.6	692	96.4	<0.0001	13	1.8	705	98.2	<0.0001	9	1.3	709	98.8	0.001
Female	696	70	10.1	626	89.9		54	7.8	642	92.2		28	4.0	668	96.0	
**Equivalent Household income**																
Fourth 25% (lowest)	418	35	8.4	383	91.6	0.271	27	6.5	391	93.5	0.136	16	3.8	402	96.2	0.318
Third 25%	394	25	6.4	369	93.7		17	4.3	377	95.7		8	2.0	386	98.0	
Second 25%	391	27	6.9	364	93.1		18	4.6	373	95.4		9	2.3	382	97.7	
Highest 25%	211	9	4.3	202	95.7		5	2.4	206	97.6		4	1.9	207	98.1	
**Education level**																
Lower than elementary school graduate	904	80	8.9	824	91.2	0.001	55	6.1	849	93.9	0.017	28	3.1	876	96.9	0.441
Middle school graduate	170	5	2.9	165	97.1		5	2.9	165	97.1		3	1.8	167	98.2	
High school graduate	233	8	3.4	225	96.6		5	2.2	228	97.9		5	2.2	228	97.9	
Upper than college graduate	107	3	2.8	104	97.2		2	1.9	105	98.1		1	0.9	106	99.1	
**Residence area**																
Urban	655	47	7.2	608	92.8	0.829	36	5.5	619	94.5	0.452	20	3.1	635	97.0	0.568
Sub-urban	356	24	6.7	332	93.3		14	3.9	342	96.1		9	2.5	347	97.5	
Rural area	403	25	6.2	378	93.8		17	4.2	386	95.8		8	2.0	395	98.0	
**Caregiver type**																
Spouse	556	27	4.9	529	95.1	0.010	16	2.9	540	97.1	0.014	7	1.3	549	98.7	0.006
Adult child or their spouse	351	37	10.5	314	89.5		25	7.1	326	92.9		16	4.6	335	95.4	
Relatives	62	3	4.8	59	95.2		3	4.8	59	95.2		3	4.8	59	95.2	
Others ^b^	22	3	13.6	19	86.4		3	13.6	19	86.4		2	9.1	20	90.9	
None	423	26	6.2	397	93.9		20	4.7	403	95.3		9	2.1	414	97.9	
**Care hour**		3 ± 5.25		3 ± 4.72			4 ± 5.36		3 ± 4.72			4 ± 6.02		3 ± 4.71		
**IADL ^c^**		5 ± 3.23		4 ± 3.23			5 ± 3.24		4 ± 3.23			6 ± 3.31		4 ± 3.23		
**BMI ^d^**		22 ± 3.14		23 ± 6.46			22 ± 3.18		23 ± 6.43			22 ± 3.26		23 ± 6.39		
**MMSE ^e^**		16 ± 8.60		18 ± 6.45			16 ± 8.70		18 ± 9.44			16 ± 9.03		18 ± 9.42		
**Number of chronic diseases**																
None	489	18	3.7	471	96.3	0.000	11	2.3	478	97.8	0.001	6	1.2	483	98.8	0.055
1	449	28	6.2	421	93.8		21	4.7	428	95.3		16	3.6	433	96.4	
2+	476	50	10.5	426	89.5		35	7.4	441	92.7		15	3.2	461	96.9	
**Regular Exercise**																
Yes	346	21	6.1	325	93.9	0.540	17	4.9	329	95.1	0.860	10	2.9	336	97.1	0.714
No	1068	75	7.0	993	93.0		50	4.7	1018	95.3		27	2.5	1041	97.5	

^a^ Number of household members living in a same household with respondents. ^b^ Caregivers who are not included in family relationship (i.e., paid caregiver, volunteers, neighbors and friends). ^c^ Higher score indicates lower level of instrumental activity of daily living (IADL) functioning. ^d^ Higher score indicates higher level of body mass index (BMI). ^e^ Higher score indicates higher level of cognitive functioning.

**Table 2 ijerph-19-05888-t002:** Odds ratios for factors associated with overall falls, falls needing medical treatment, and hip fractures caused by falls.

Variable	Overall Falls			Falls Needing Medical Treatment			Hip Fractures Caused by Falls		
	OR	95% CI		OR	95% CI		OR	95% CI	
**Number of household members ^a^**									
0 (single household)	2.13	1.36	3.34	2.13	1.28	3.53	1.93	1.01	3.69
1	1.54	1.08	2.19	1.41	0.95	2.09	1.58	0.98	2.53
2≤	1.00			1.00			1.00		
**Age**									
45–64	1.19	0.61	2.31	1.42	0.69	2.91	0.85	0.34	2.10
≤75	1.67	0.95	2.92	1.56	0.84	2.90	1.03	0.48	2.23
≤85	1.39	0.85	2.27	1.14	0.66	1.95	0.87	0.45	1.69
85<	1.00			1.00			1.00		
**Sex**									
Male	0.38	0.25	0.55	0.27	0.17	0.43	0.31	0.17	0.55
Female	1.00			1.00			1.00		
**Equivalent Household income**									
Fourth 25% (lowest)	1.18	0.71	1.98	1.48	0.84	2.63	1.27	0.66	2.46
Third 25%	0.82	0.50	1.35	0.96	0.54	1.69	0.78	0.42	1.47
Second 25%	1.10	0.68	1.76	1.16	0.67	2.01	0.85	0.44	1.65
Highest 25%	1.00			1.00			1.00		
**Education level**									
Lower than elementary school graduate	1.22	0.55	2.71	0.84	0.35	2.02	0.51	0.18	1.41
Middle school graduate	1.10	0.47	2.59	0.99	0.38	2.56	0.46	0.14	1.49
High school graduate	1.03	0.45	2.37	0.78	0.30	2.00	0.66	0.24	1.83
Upper than college graduate	1.00			1.00			1.00		
**Residence area**									
Urban	1.20	0.83	1.74	1.23	0.81	1.88	1.01	0.59	1.73
Sub-urban	1.26	0.85	1.85	1.07	0.68	1.67	1.20	0.69	2.08
Rural area	1.00			1.00			1.00		
**Caregiver type**									
Spouse	1.10	0.70	1.71	1.11	0.66	1.88	0.96	0.46	1.98
Family except for spouse	1.10	0.72	1.67	1.16	0.73	1.83	1.19	0.63	2.24
Others ^b^	1.15	0.56	2.36	1.22	0.59	2.53	0.95	0.38	2.39
None	1.00			1.00			1.00		
**Care hour**	0.99	0.96	1.03	1.00	0.97	1.04	1.02	0.98	1.06
**IADL ^c^**	1.09	1.03	1.15	1.07	1.01	1.14	1.11	1.02	1.21
**BMI ^d^**	0.94	0.90	0.98	0.93	0.89	0.98	0.96	0.90	1.02
**MMSE ^e^**	1.02	0.99	1.04	1.01	0.99	1.03	1.03	1.00	1.06
**Number of chronic diseases**									
None	0.98	0.63	1.53	0.98	0.60	1.60	1.14	0.60	2.14
1	1.39	0.89	2.17	1.28	0.78	2.10	1.45	0.74	2.84
2+	1.00			1.00			1.00		
**Regular Exercise**									
Yes	0.88	0.61	1.28	0.88	0.56	1.37	0.69	0.36	1.32
No	1.00			1.00			1.00		
**Year**									
2006	1.89	1.23	2.90	1.54	0.94	2.53	1.47	0.79	2.72
2008	1.18	0.76	1.85	1.29	0.78	2.15	1.22	0.65	2.30
2010	1.03	0.66	1.60	0.88	0.52	1.49	1.03	0.53	2.01
2012	1.00			1.00			1.00		

^a^ Number of household members living in the same household as respondents. ^b^ Caregivers who are not included in family relationship (i.e., paid caregiver, volunteers, neighbors and friends). ^c^ Higher score indicates lower level of instrumental activity of daily living (IADL) functioning. ^d^ Higher score indicates higher level of body mass index (BMI). ^e^ Higher score indicates higher level of cognitive functioning.

**Table 3 ijerph-19-05888-t003:** Odds ratios for occurrence of overall falls, falls needing medical treatment and falls resulting in hip fractures according to number of family members in household according to cognitive function.

Variable	Overall Falls			Falls Needing Medical Treatment			Hip Fractures Caused by Falls		
	OR ^†^	95% CI		OR ^†^	95% CI		OR ^†^	95% CI	
**Cognitive decline ^a^**									
**Number of household members ^b^**									
0 (single household)	2.08	1.27	3.38	2.15	1.24	3.72	2.15	1.05	4.43
1	1.44	0.98	2.13	1.23	0.79	1.94	1.42	0.81	2.50
2≤	1.00			1.00			1.00		
**Normal congnitive function**									
**Number of household members ^b^**									
0 (single household)	1.22	0.32	4.64	0.99	0.18	5.55	0.72	0.11	4.79
1	1.21	0.51	2.85	1.55	0.48	4.99	1.94	0.75	5.03
2≤	1.00			1.00			1.00		

^a^ Cognitive decline includes those with cognitive decline and cognitive impairment based on their MMSE score (MMSE score ≥ 18). ^b^ Number of household members living in the same household as respondents. ^†^ Adjusted for age, sex, equivalent household income, education level, residence area, caregiver type, care hour, IADL, BMI, number of chronic diseases, and regular exercise.

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
