# Peer review of "Effect of Number of Household Members on Falls among Disabled Older People"

_ijerph, 2022, doi:10.3390/ijerph19105888_

Round 1

Reviewer 1 Report

Overall recommendation: Accept with minor changes

Please add the following:

  1. A simple figure in the form of a flow chart explaining all the selection and exclusion criteria, various parameters considered and statistical tests performed.

  1. Do you think there would be a difference if you added another category to the number of households in the category – instead of just none, on and two or more does the analysis change when you categorise as following: None, one, two, three, four or more? If not – need not include it. If it does – you can decide based on the results if you wish to add it. I’ll leave it to you.

Author Response

Reviewer 1

  1. A simple figure in the form of a flow chart explaining all the selection and exclusion criteria, various parameters considered and statistical tests performed.

Reply and correction: Thank you for your comment. As you suggested, we added a flowchart to explain our inclusion/exclusion criteria. The flowchart was added in p.13 as Figure 1.

Also, we revised the sentences regarding various variables and statistical methods in this study in Methods section. We tried to give better information about the information about selection criteria and methods we used. We would like to request your kind understanding. If you think it is not enough, please tell us.

  1. Do you think there would be a difference if you added another category to the number of households in the category – instead of just none, on and two or more does the analysis change when you categorise as following: None, one, two, three, four or more? If not – need not include it. If it does – you can decide based on the results if you wish to add it. I’ll leave it to you.

Reply and correction: Thank you for your comment. I tried analyzing the association between the number of households and falls by switching to the category as you suggested (None, one, two, three, four or more). However, we found that the results is insignificant and the association is similar. In this situation, we think that maintaining the original category we firstly used (None, one, two or more) would be enough to show the association between the number of households and falls. Therefore, we decided to maintain the original category we firstly used (None, one, two or more). We would like to request your kind understanding.

Reviewer 2 Report

The paper entitled “Effect of Number of Household Members on Falls among Disabled Older People” addresses an important topic in geriatric research regarding fall (prevention). The main purpose is to investigate an unusual topic, the number of cohabitating household members, on falls among disabled older people. 

The main strengths include the large sample size of more than 1000 participants.

 The conceptual idea needs to be reframed.
“The risk factors for falls have been reported to include balance deficits, living arrangement, history of previous falls, difficulties with ambulation, visual impairments, age, gender, having multiple risk factors, as well as certain chronic diseases (Schiller, Kramarow, & Dey, 2007).” – please upgrade this reference. There is more recent evidence. 
What do authors mean by living arrangements? It needs to be clarified. 

 Why living alone per se should predict falls? It is not clear in the introduction. 
The authors say that “Although the existence of a household member can be important for older persons with a disability to reduce falls and injuries caused due to falls, studies examining the association between the number of household members and falls, and injuries caused due to falls are insufficient, especially those on the disabled older population”. Is there any evidence supporting this idea? 

 Probably people at fall risk are particularly frail individuals that have special needs.

 Perhaps those individuals need assistance for most of the activities of daily living.

 Considering this issue in the analyses should eliminate the direct relationship observed.

 Besides that, it is unlikely that there are people available to assist all the time in most households.
 Perhaps a study design comparing with geriatric care may help.

 What does the study suggest regarding the prevention of falls?
 Is it the same for people residing at home and in an institution?
 This issue needs to be elaborated on in more detail.

 Thus, the conceptual and empirical approach, together with practical
 implications need to be better justified.

The references used need to be upgraded. there are more recent papers on the topic that was not cited. 

Author Response

Reviewer 2

The paper entitled “Effect of Number of Household Members on Falls among Disabled Older People” addresses an important topic in geriatric research regarding fall (prevention). The main purpose is to investigate an unusual topic, the number of cohabitating household members, on falls among disabled older people. The main strengths include the large sample size of more than 1000 participants.

1) The conceptual idea needs to be reframed.

Reply: Thank you for your comment. We have revised the sentences throughout Introduction section.

2) “The risk factors for falls have been reported to include balance deficits, living arrangement, history of previous falls, difficulties with ambulation, visual impairments, age, gender, having multiple risk factors, as well as certain chronic diseases (Schiller, Kramarow, & Dey, 2007).” – please upgrade this reference. There is more recent evidence. 
Reply and correction: Thank you for your comment. We have changed to more recent evidence.

3) What do authors mean by living arrangements? It needs to be clarified. 
Reply and correction: In previous studies, living arrangements include the number of household member or the type of family member. The factors associated with living arrangements has known to have implications for the well-being of older adults, and there have been many studies to understanding the determinants of the survival and functioning of older populations regarding this point (Martin and Kinsella 1994).

In this study, we especially focused on the number of household of older adults among factors related to living arrangement. We revised the sentence regarding this point as follows: “Therefore, focusing on characteristics related to their living arrangement such as the number or composition of households is important to understand and plan for proper care of these individuals [10].”

4) Why living alone per se should predict falls? It is not clear in the introduction. 
Reply and correction: Then number of household member living with disable older people related to the amount of supportive care from their household members to prevent fall. Considering that those disabled older adults living alone who have difficulty to moving and daily activities, living alone could be associated with higher falls than those living with more household members. We have changed the sentences regarding this point in the 3rd paragraph of Introduction section of p.2. as follows: “The number of household members residing with those older adults might be associated with falls. Because the larger number of family member for older adults with functional disability is found to be associated with better instrumental and social support [11], and the amount of receiving any type of care including formal and informal care from cohabitating household members can be small or none for those older adults living alone [12,13]. In addition, the number of household members is known to be related to reduce the survival and functioning decline of older populations [10].”

5) The authors say that “Although the existence of a household member can be important for older persons with a disability to reduce falls and injuries caused due to falls, studies examining the association between the number of household members and falls, and injuries caused due to falls are insufficient, especially those on the disabled older population”. Is there any evidence supporting this idea? 
Reply and correction: This study investigated to the association between the number of household members and falls and injuries due to falls. When we searched for relevant prior studies, previous studies mostly focused on the status of cohabitating, but no studies on falls according to the number of families. Also, the existence of a household member can be important for older persons with a disability to reduce falls and injuries caused due to falls, studies examining the association between number of household members and falls, and injuries caused due to falls are insufficient, especially those on the disabled older population. Finally, studies regarding this issue have not been conducted in Korea, which is one of the most rapidly aging developed countries, despite the importance of this issue among the disabled aging population. Although it is necessary to study the relationship between the number of household members and falls to identify risk factors associated with falls for fall management of elderly households with disabilities who have difficulties in performing daily life, there has been no such study.

6) Probably people at fall risk are particularly frail individuals that have special needs. Perhaps those individuals need assistance for most of the activities of daily living. Considering this issue in the analyses should eliminate the direct relationship observed.
Reply and correction: We agree to your opinion. As you mentioned, we also thought that frail individuals could have special needs or assistance for activities of daily living. This study was conducted based on the data from Korea Longitudinal Study of Aging (KLoSA) which is secondary data. Therefore, we tried to control those factors as possible as we can by adding some possibly accessible information in KLoSA dataset. We added variables including care type, care hour, IADL, MMSE to control in our statistical model. However, we understand that there might be remaining potential confounding. Therefore, we added this point as one of our limitations. We revised the sentence regarding this point in 4th paragraph of Discussion section of p.6. as follows: However, there are some limitations in this study. First, the longitudinal study design used in this study diminishes our ability to determine the causal relationship between number of household members, and falls and injuries caused due to falls. In other words, our results could reflect a causal relationship contrary to that ascribed to the relationship between number of household members, and falls and injuries caused due to falls (e.g., experience of fall or injuries due to falls leads to living alone). Future studies might inves-tigate the effect of changes in the number of household members on health outcomes or the effect of the same on later health outcomes. Secondly, respondents’ reports were sub-jective and imperfect measures, potentially affected by a perception bias and adaptation of resources. Finally, on account of the observational nature of this study, this study cannot completely eliminate the possibility for residual confounding and other potential sources of bias.”

7) Besides that, it is unlikely that there are people available to assist all the time in most households. Perhaps a study design comparing with geriatric care may help.
Reply and correction: This study was performed based on the Korean population 45 years or more by KLoSA, and the KLoSA survey was done only for those Korean population 45 years or more living in home. Therefore, the information about receiving geriatric care or living in geriatric care center is not included in KLoSA dataset. We are sorry to say this, but it is not possible to do comparing with geriatric care. We will add this point as one of our limitations. We revised the sentence regarding this point in 4th paragraph of Discussion section of p.6. as follows: “Finally, on account of the observational nature of this study, this study cannot completely eliminate the possibility for residual confounding such as receiving geriatric care and other potential sources of bias.”

8) What does the study suggest regarding the prevention of falls? Is it the same for people residing at home and in an institution? This issue needs to be elaborated on in more detail.
Reply and correction: This study is only focused on those people living at home. By studying the association between the number of household members and falls among Korean disabled population, this study shows that those disabled people who are one of the most vulnerable population of fall is related to living alone. This result of this study addressed that those disabled people living alone is the most vulnerable population of fall, and proper support needed to prevent and manage falls by approaching the association between the number of household members and falls for those disabled people living at home. We revised the sentence regarding this point in 4th paragraph of Discussion section of p.6. as follows: In addition, the association between the number of household members and falls can be dif-fered by older adults’ residential place such as home and institution. However, this study only focused on those people living at home because KLoSA survey is targeted to only for those indi-viduals living at home. Further studies about this study topic need to compare with those living in an institution.”

 9) Thus, the conceptual and empirical approach, together with practical implications need to be better justified.
Reply: Thank you for your opinion. We revised the sentences throughout the whole part of Introduction and Conclusion section to give better both conceptual and implications.

10) The references used need to be upgraded. there are more recent papers on the topic that was not cited. 
Reply: Thank you for your opinion. We have changed to more recent studies.

Reviewer 3 Report

Thank you for the opportunity to review this interesting article aimed to investigate the effects of number of cohabitating household members on falls among the disabled aging Korean population.
The article is of scientific interest and in line with the aims of the journal. The author's guidelines have been respected and the overall structure of the manuscript is good and does not require a revision of the English language. I recommend publication after minor revision.

INTRODUCTION
The introduction argues well the topic covered. I suggest summarizing the introduction so as to make it more accessible and to highlight the aims of the study.
The aim of the present article must be rephrased and be more clear for the readers.
MATERIALS AND METHODS
The materials and methods section adequately describes the authors' work.
The inclusion and exclusion criteria could be better explained.
RESULTS AND DISCUSSION
The results are written in a very fluid and readable way for the readers.
The discussion correctly argues the results identified.
CONCLUSIONS
Please develop the conclusion section further

Author Response

Reviewer 3

Thank you for the opportunity to review this interesting article aimed to investigate the effects of number of cohabitating household members on falls among the disabled aging Korean population.
The article is of scientific interest and in line with the aims of the journal. The author's guidelines have been respected and the overall structure of the manuscript is good and does not require a revision of the English language. I recommend publication after minor revision.

1) INTRODUCTION
The introduction argues well the topic covered. I suggest summarizing the introduction so as to make it more accessible and to highlight the aims of the study. The aim of the present article must be rephrased and be more clear for the readers.
Reply: Thank you for your opinion. We have revised the sentences throughout the whole Introduction section, and we tried to summarize the aims of this study.

2) MATERIALS AND METHODS
The materials and methods section adequately describes the authors' work.
The inclusion and exclusion criteria could be better explained.
Reply and correction: The study population of this study is individuals aged 45 years and older who need any help for performance of either activities of daily living (ADL) or instrumental activities of daily living (IADL). Therefore, we included only those participants who did not need any help either for ADL or IADL. Among them, we excluded those participants who had missing information needed for this study. To give better information, we decided to revise some sentences regarding this point in the sentences in the 3rd paragraph of Methods section of p.3. The revised sentences are as follows: “This study targeted individuals who need any help for performance in either activities of daily living (ADL) or instrumental activities of daily living (IADL). Therefore, 1,516 participants among the 10,254 participants included in the 2006 survey (baseline year), those who did not need any help for either ADL or IADL (n= 8,738) were excluded from the study population. In addition, we only included those participants who had all the information needed for this study.“

3) RESULTS AND DISCUSSION
The results are written in a very fluid and readable way for the readers.
The discussion correctly argues the results identified.
Reply: Thank you for your comment.

4) CONCLUSIONS
Please develop the conclusion section further
Reply and correction: Thank you for your comment. As you requested, we revised the sentences in Conclusion section in p.7. as follows: “Our results indicate that living alone is associated with higher odds of overall falls, falls needing medical treatment, and hip fracture caused by falls, particularly in those with cognitive decline. Considering that those living alone were identified to be a vulnera-ble group of fall, more effective fall prevention intervention should be designed. Especially, governmental support need to prevent and manage care for preventing falls in disabled older adults living alone by encouraging to use home visiting care services and promoting the use of safety equipment.”

Reviewer 4 Report

This paper investigated the Effect of Number of Household Members on Falls among Disabled Older People. Generally, this is a well written and well substantiated paper. However, the organization of the contents needs a bit of work. I hope my comments will help improving your manuscript.

1. Please provide a clear reason for a fall survey targeting the 45 over age.

2. Regarding the question about falls, do you not survey the number of falls as well as "yes" or "no" for the past two years?

3. Please describe the inclusion and exclusion criteria.

4. Did middle-aged and older adults living alone have the same high rate of falls as older adults?

5. Can you present data on the frequency of going out for this subject?

Author Response

Reviewer 4

This paper investigated the Effect of Number of Household Members on Falls among Disabled Older People. Generally, this is a well written and well substantiated paper. However, the organization of the contents needs a bit of work. I hope my comments will help improving your manuscript.

  1. Please provide a clear reason for a fall survey targeting the 45 over age.
    Reply and correction: Since the influence of factors on falls in the elderly occurs from middle age, we thought that it was meaningful to include from middle age. Similarly, since there are studies that included middle-aged people in the previous related studies, they were included from the age of 45 or older. In addition, since the data we used also includes the middle-aged, we selected as a target 45 years of age or older, thinking that it is meaningful to comprehensively understand factors that can be a risk to the health of the elderly, including all from middle-aged. We would like to request your kind understanding.

  1. Regarding the question about falls, do you not survey the number of falls as well as "yes" or "no" for the past two years?
    Reply and correction: This study was conducted based on the data from Korea Longitudinal Study of Aging (KLoSA) which is secondary data. When we checked KLoSA dataset, information about the number of falls was available. We tried to check if there is an association between the number of household members and the number of falls. However, we couldn’t find any statistically significant association between the number of household members and the number of falls. We would like to request your kind understanding.

Table. Adjusted effect of the number of household members on the number of falls

Variable

Number of falls

β

SE

P-value

Number of household membersa

0 (single household)

-0.005

0.151

0.972

1

-0.035

0.110

0.752

2≤

Ref.

  1. Please describe the inclusion and exclusion criteria.
    Reply and correction: The study population of this study is individuals aged 45 years and older who need any help for performance of either activities of daily living (ADL) or instrumental activities of daily living (IADL). Therefore, we included only those participants who did not need any help either for ADL or IADL. Among them, we excluded those participants who had missing information needed for this study. To give better information, we decided to revise some sentences regarding this point in the sentences in the 3rd paragraph of Methods section of p.3. The revised sentences are as follows: “This study targeted individuals who need any help for performance in either activities of daily living (ADL) or instrumental activities of daily living (IADL). Therefore, 1,516 participants among the 10,254 participants included in the 2006 survey (baseline year), those who did not need any help for either ADL or IADL (n= 8,738) were excluded from the study population. In addition, we only included those participants who had all the information needed for this study. Therefore, we excluded participants with missing information on Body Mass Index (BMI) (n = 98) and care hours (n = 4). Thus, the final sample at 2006 comprised 1,414 observations. The number of observations of follow-up years was as fol-lows: 1063 in 2008 survey, 995 in 2010 survey, and 845 in 2012 survey, respectively (Figure 1).”

  1. Did middle-aged and older adults living alone have the same high rate of falls as older adults?
    Reply: As middle-aged and elderly people can show different characteristics, we analyzed to see if middle-aged and elderly living alone had as high a fall rate as the elderly. When we checked the dataset, those people living alone with fall were found 1 (0.07%) in mid-aged (45 to 64), and 15 (1.06%) among those aged 65 or more. The rate of fall was higher among those aged 65 or more than those middle aged.

  1. Can you present data on the frequency of going out for this subject?
    Reply and correction: As we mentioned above, we included only those participants who did not need any help either for ADL or IADL. Among them, we excluded those participants who had missing information needed for this study. Firstly, 1,516 participants included among the 10,254 participants in the 2006 survey (baseline year), those who did not need any help for either ADL or IADL (n= 8,738) were excluded from the study population. Secondly, we only included those participants who had all the information needed for this study. Therefore, we excluded participants with missing information on Body Mass Index (BMI) (n = 98) and care hours (n = 4). Thus, the final sample at 2006 comprised 1,414 observations. The number of observations of follow-up years was as follows: 1063 in 2008 survey, 995 in 2010 survey, and 845 in 2012 survey, respectively (Figure 1).We added a flowchart in p.13. to present the study population (Figure 1).

Also, we added the sentences regarding this point in the 3rd paragraph of Methods section of p.3. as follows: “Therefore, 1,516 participants among the 10,254 participants included in the 2006 survey (baseline year), those who did not need any help for either ADL or IADL (n= 8,738) were excluded from the study population. We only included those participants who had all the information needed for this study. Therefore, we excluded participants with missing information on Body Mass Index (BMI) (n = 98) and care hours (n = 4). Thus, the final sample at 2006 comprised 1,414 observations. The number of observations of follow-up years was as fol-lows: 1063 in 2008 survey, 995 in 2010 survey, and 845 in 2012 survey, respectively (Figure 1).”

We would like to thank you for your consideration of our revised manuscript for publication.  We look forward to a positive response.

Round 2

Reviewer 2 Report

I don't have further comments on the paper.